# Evidence for embracing normative modeling

Saige Rutherford[1,2,3]*, Pieter Barkema[2], Ivy F Tso[3,4], Chandra Sripada[3,5], Christian F Beckmann[1,2,6†], Henricus G Ruhe[2,7†], Andre F Marquand[1,2†]

[1]Department of Cognitive Neuroscience, Radboud University Nijmegen Medical Centre, Nijmegen, Netherlands; [2]Donders Institute, Radboud University Nijmegen, Nijmegen, Netherlands; [3]Department of Psychiatry, University of Michigan-Ann Arbor, Ann Arbor, United States; [4]Department of Psychology, University of Michigan-Ann Arbor, Ann Arbor, United States; [5]Department of Philosophy, University of Michigan-Ann Arbor, Ann Arbor, United States; [6]Center for Functional MRI of the Brain (FMRIB), Nuffield Department for Clinical Neuroscience, Welcome Centre for Integrative Neuroimaging, Oxford University, Oxford, United Kingdom; [7]Department of Psychiatry, Radboud University Nijmegen Medical Centre, Nijmegen, Netherlands

**Abstract** In this work, we expand the normative model repository introduced in Rutherford et al., 2022a to include normative models charting lifespan trajectories of structural surface area and brain functional connectivity, measured using two unique resting-state network atlases (Yeo-17 and Smith-10), and an updated online platform for transferring these models to new data sources. We showcase the value of these models with a head-to-head comparison between the features output by normative modeling and raw data features in several benchmarking tasks: mass univariate group difference testing (schizophrenia versus control), classification (schizophrenia versus control), and regression (predicting general cognitive ability). Across all benchmarks, we show the advantage of using normative modeling features, with the strongest statistically significant results demonstrated in the group difference testing and classification tasks. We intend for these accessible resources to facilitate the wider adoption of normative modeling across the neuroimaging community.

*For correspondence:
saige.rutherford@donders.ru.nl

†These authors contributed equally to this work

## Editor's evaluation

This is a rigorous and compelling extension of previous normative modeling work. The current study demonstrates that normative models incorporating lifespan trajectories of structural and functional connectivity provide a strong basis for brain imaging studies across a range of tasks including, univariate group difference assessment, classification, and building regression models. The work is important, rigorous and a valuable contribution to the field.

## Introduction

Normative modeling is a framework for mapping population-level trajectories of the relationships between health-related variables while simultaneously preserving individual-level information (*Marquand et al., 2016a*; *Marquand et al., 2016b*; *Rutherford et al., 2022b*). Health-related variables is an intentionally inclusive and broad definition that may involve demographics (i.e. age and gender), simple (i.e. height and weight), or complex (i.e. brain structure and function, genetics) biological measures, environmental factors (i.e. urbanicity, pollution), self-report measures (i.e. social satisfaction, emotional experiences), or behavioral tests (i.e. cognitive ability, spatial reasoning). Charting the relationships, as mappings between a covariate (e.g. age) and response variable (e.g.

brain measure) in a reference population creates a coordinate system that defines the units in which humans vary. Placing individuals into this coordinate system creates the opportunity to characterize their profiles of deviation. While this is an important aspect of normative modeling, it is usually just the first step, i.e., you are often interested in using the outputs of normative models in downstream analyses to detect case-control differences, stratification, or individual statistics. This framework provides a platform for such analyses as it effectively translates diverse data to a consistent scale, defined with respect to population norms.

Normative modeling has seen widespread use spanning diverse disciplines. The most well-known example can be found in pediatric medicine, where conventional growth charts are used to map the height, weight, and head circumference trajectories of children (*Borghi et al., 2006*). Under the neuroscience umbrella, generalizations of this approach have been applied in the fields of psychiatry (*Floris et al., 2021*; *Madre et al., 2020*; *Wolfers et al., 2015*; *Wolfers et al., 2017*; *Wolfers et al., 2018*; *Wolfers et al., 2020*; *Wolfers et al., 2021*; *Zabihi et al., 2019*; *Zabihi et al., 2020*), neurology (*Itälinna et al., 2022*; *Verdi et al., 2021*), developmental psychology (*Holz et al., 2022*; *Kjelkenes et al., 2022*), and cognitive neuroscience (*Marquand et al., 2017*). Throughout these numerous applications, normative models have exposed the shortcomings of prior case-control frameworks, i.e., that they rely heavily on the assumption, there is within-group homogeneity. This case versus control assumption is often an oversimplification, particularly in psychiatric diagnostic categories, where the clinical labels used to place individuals into group categories are often unreliable, poorly measured, and may not map cleanly onto underlying biological mechanisms (*Cai et al., 2020*; *Cuthbert and Insel, 2013*; *Flake and Fried, 2020*; *Insel et al., 2010*; *Linden, 2012*; *Loth et al., 2021*; *Michelini et al., 2021*; *Moriarity et al., 2022*; *Moriarity and Alloy, 2021*; *Nour et al., 2022*; *Sanislow, 2020*; *Zhang et al., 2021*). Correspondingly, traditional analysis techniques for modeling case versus control effects have often led to null findings (*Winter et al., 2022*) or significant but very small clinically meaningless differences. These effects are furthermore frequently aspecific to an illness or disorder (*Baker et al., 2019*; *Goodkind et al., 2015*; *McTeague et al., 2017*; *Sprooten et al., 2017*) and inconsistent or contradictory (*Filip et al., 2022*; *Lee et al., 2007*; *Pereira-Sanchez and Castellanos, 2021*) yielding questionable clinical utility (*Etkin, 2019*; *Mottron and Bzdok, 2022*).

In addition to the applications of normative modeling, there is also active technical development (*Dinga et al., 2021*; *Fraza et al., 2022*; *Fraza et al., 2021*; *Kia et al., 2020*; *Kia et al., 2021*; *Kia and Marquand, 2018*; *Kumar, 2021*). Due to the growing popularity of normative modeling and in recognition of the interdisciplinary requirements using and developing this technology (clinical domain knowledge, statistical expertise, data management, and computational demands), research interests have been centered on open science, and inclusive, values (*Gau et al., 2021*; *Levitis et al., 2021*) that support this type of interdisciplinary scientific work. These values encompass open-source software, sharing pre-trained big data models (*Rutherford et al., 2022a*), online platforms for communication and collaboration, extensive documentation, code tutorials, and protocol-style publications (*Rutherford et al., 2022b*).

The central contribution of this paper is to, first, augment the models in *Rutherford et al., 2022a*, with additional normative models for surface area and functional connectivity, which are made open and accessible to the community. Second, we comprehensively evaluate the utility of normative models for a range of downstream analyses, including (1) mass univariate group difference testing (schizophrenia versus controls), (2) multivariate prediction – classification (using support vector machines to distinguish schizophrenia from controls), and (3) multivariate prediction – regression (using principal component regression (PCR) to predict general cognitive ability) (*Figure 1*). Within these benchmarking tasks, we show the benefit of using normative modeling features compared to using raw features. We aim for these benchmarking results, along with our publicly available resources (code, documentation, tutorials, protocols, community forum, and website for running models without using any code). Combined this provides practical utility as well as scientific evidence for embracing normative modeling.

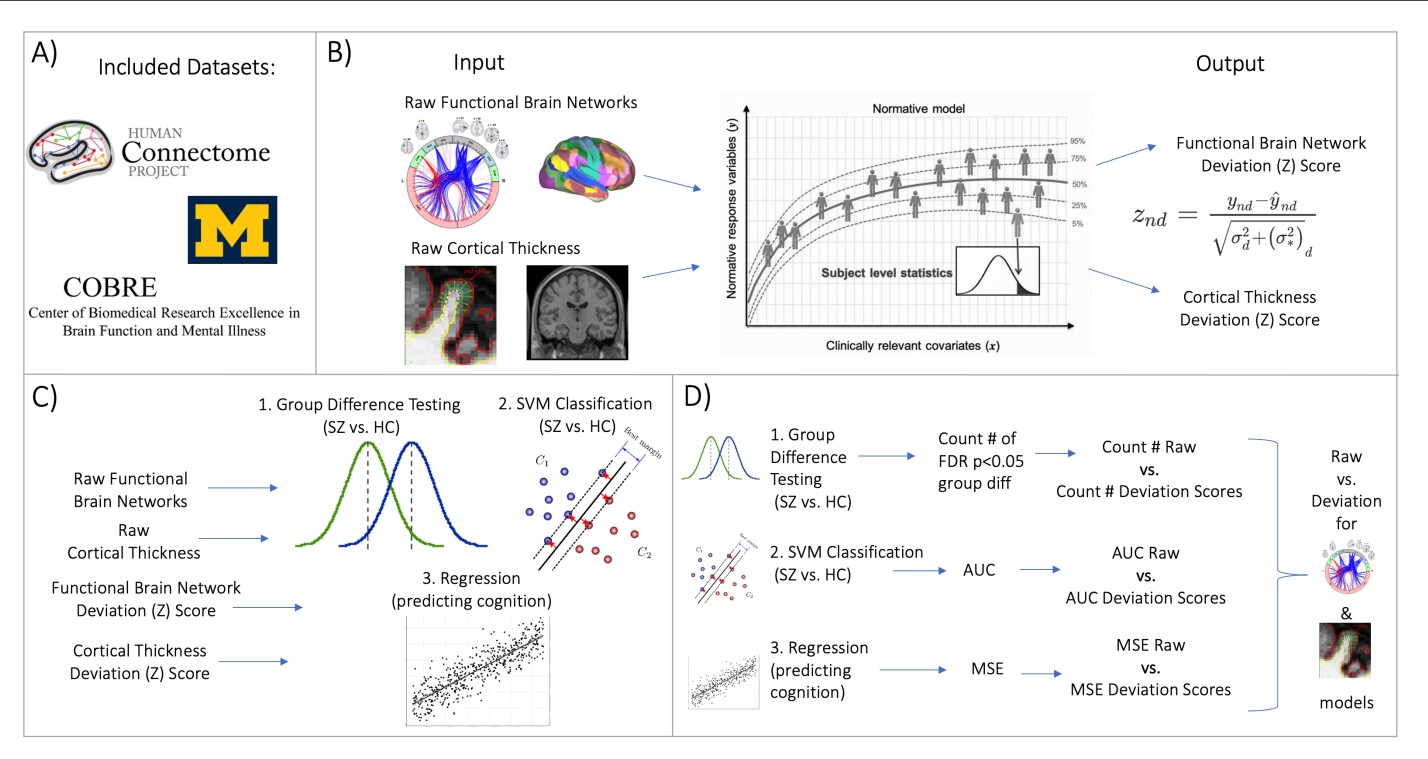

**Figure 1.** Overview of workflow. (**A**) Datasets included the Human Connectome Project (young adult) study, the University of Michigan schizophrenia study, and the Center for Biomedical Research Excellence (COBRE) schizophrenia study. (**B**) Openly shared, pre-trained on big data, normative models were estimated for large-scale resting-state functional brain networks and cortical thickness. (**C**) Deviation (Z) scores and raw data, for both functional and structural data, were input into three benchmarking tasks: 1. group difference testing, 2. support vector machine (SVM) classification, and 3. regression (predicting cognition). (**D**) Evaluation metrics were calculated for each benchmarking task. These metrics were calculated for the raw data models and the deviation score models. The difference between each models' performance was calculated for both functional and structural modalities.

## Methods
### Dataset selection and scanner parameters

Datasets used for training the functional normative models closely match the sample included in *Rutherford et al., 2022a*, apart from sites that did not collect or were unable to share functional data. Evaluation of the functional normative models was performed in a test set (20% of the training set) and in two transfer sets that are comprised of scanning sites not seen by the model during training (clinical and healthy controls). The full details of the data included in the functional normative model training can be found in Appendix 1 and *Supplementary file 1*. We leverage several datasets for the benchmarking tasks, the Human Connectome Project Young Adult study (HCP) (*Van Essen et al., 2013*), The Center for Biomedical Research Excellence (COBRE) (*Aine et al., 2017*; *Sui et al., 2018*), and the University of Michigan SchizGaze (UMich) (*Tso et al., 2021*; *Table 1*). The HCP data was chosen because it is widely used by the neuroscience community, especially for prediction studies. Also, prior studies using HCP data have shown promising results for predicting general cognitive ability (*Sripada*

**Table 1.** Dataset inclusion and sample overview.

| Study | Benchmark Task | Cortical Thickness | | | Functional Networks | | |
|---|---|---|---|---|---|---|---|
| | | N | Age (m, s.d.) | F, M (%) | N | Age (m, s.d.) | F, M (%) |
| HCP | Regression – predicting cognition | 529 | 28.8, 3.6 | 53.4, 46.6 | 499 | 28.9, 3.6 | 54.3, 45.6 |
| COBRE | Classification & Group Difference | 124 | 37.0, 12.7 | 24.2, 75.8 | 121 | 35.4, 12.4 | 23.1, 76.9 |
| UMich | Classification & Group Difference | 89 | 32.6, 9.6 | 50.6, 49.3 | 87 | 33.0, 10.1 | 50.6, 49.3 |

*et al., 2020a*). The HCP data was used in the prediction – regression benchmarking task. The COBRE and UMich datasets are used in the classification and group difference testing benchmarking tasks. Inclusion criteria across all the datasets were that the participant has necessary behavioral and demographic variables, as well as high-quality MRI data. High-quality was defined for structural images as in our prior work (*Rutherford et al., 2022a*), namely as the lack of any artifacts such as ghosting or ringing, that Freesurfer surface reconstruction was able to run successfully, and that the Euler number calculated from Freesurfer (*Klapwijk et al., 2019*), which is a proxy metric for scan quality, was below a chosen threshold (rescaled Euler <10) (*Kia et al., 2022*). High-quality functional data followed recommended practices (*Siegel et al., 2017*) and was defined as having a high-quality structural MRI (required for co-registration and normalization) and at least 5 min of low motion data (framewise displacement <0.5 mm). The HCP, COBRE, and UMich functional and structural data were manually inspected for quality at several tasks during preprocessing (after co-registration of functional and structural data and after normalization of functional data to MNI template space).

All subjects provided informed consent. Subject recruitment procedures and informed consent forms, including consent to share de-identified data, were approved by the corresponding university institutional review board where data were collected. The scanning acquisition parameters were similar but varied slightly across the studies, details in Appendix 1.

## Demographic, cognition, and clinical diagnosis variables

Demographic variables included age, sex, and MRI scanner site. A latent variable of cognition, referred to as General Cognitive Ability (GCA), was created for the regression benchmarking task using HCP data. The HCP study administered the NIHToolbox Cognition battery (*Gershon et al., 2010*), and a bi-factor model was fit (for further modeling details and assessment of model fit see *Sripada et al., 2020b*). For COBRE and UMich studies, clinical diagnosis of schizophrenia was confirmed using the Structured Clinical Interview used for DSM-5 disorders (SCID) (*First, 1956*). All subjects were screened and excluded if they had: a history of neurological disorder, mental retardation, severe head trauma, or substance abuse/dependence within the last 6 (UMich) or 12 months (COBRE), were pregnant/nursing (UMich), or had any contraindications for MRI.

## Image preprocessing

Structural MRI data were preprocessed using the Freesurfer (version 6.0) recon-all pipeline (*Dale et al., 1999*; *Fischl et al., 2002*; *Fischl and Dale, 2000*) to reconstruct surface representations of the volumetric data. Estimates of cortical thickness and subcortical volume were then extracted (aparc and aseg) for each subject from their Freesurfer output folder, then merged, and formatted into a csv file (rows = subjects, columns = brain ROIs). We also share models of surface area, extracted in the same manner as the cortical thickness data from a similar dataset (described in *Supplementary file 2*).

Resting-state data were preprocessed separately for each study using fMRIPrep *Esteban et al., 2019*; however, similar steps were done to all resting-state data following best practices including field-map correction of multi-band data, slice time correction (non-multi-band data), co-registration of functional to structural data, normalization to MNI template space, spatial smoothing (2 x voxel size, 4–6 mm), and regression of nuisance confounders (WM/CSF signals, non-aggressive AROMA components [*Pruim et al., 2015a*; *Pruim et al., 2015b*], linear and quadratic effects of motion).

Large-scale brain networks from the 17-network Yeo atlas (*Yeo et al., 2011*) were then extracted and between-network connectivity was calculated using full correlation. We also shared functional normative models using the Smith-10 ICA-based parcellation (*Smith et al., 2009*) which includes subcortical coverage, however, the benchmarking tasks only use the Yeo-17 functional data. Fisher r-to-z transformation was performed on the correlation matrices. If there were multiple functional runs, connectivity matrices were calculated separately for each run then all runs for a subject were averaged. For further details regarding the preparation of the functional MRI data, see Appendix 1.

## Normative model formulation

After dataset selection and preprocessing, normative models were estimated using the Predictive Clinical Neuroscience toolkit (PCNtoolkit), an open-source python package for normative modeling (*Marquand et al., 2021*). For the structural data, we used a publicly shared repository of pre-trained normative models that were estimated on approximately 58,000 subjects using a warped Bayesian

Linear Regression algorithm (*Fraza et al., 2021*). The covariates used to train the structural normative models included age, sex, data quality metric (Euler number), and site. Normative models of surface area were also added to the same repository *Supplementary file 2*. Model fit was established using explained variance, mean standardized log loss, skew, and kurtosis. The outputs of normative modeling also include a Z-score, or deviation score, for all brain regions and all subjects. The deviation score represents where the individual is in comparison to the population the model was estimated on, where a positive deviation score corresponds to the greater cortical thickness or subcortical volume than average, and a negative deviation score represents less cortical thickness or subcortical volume than average. The deviation (Z) scores that are output from the normative model are the features input for the normative modeling data in the benchmarking analyses.

In addition to normative models of brain structure, we also expanded our repository by estimating normative models of brain functional connectivity (resting-state brain networks, Yeo-17 and Smith-10) using the same algorithm (Bayesian Linear Regression) as the structural models. The covariates used to train the functional normative models were similar to the structural normative models which included age, sex, data quality metric (mean framewise displacement), and site. Functional normative models were trained on a large multi-site dataset (approx. N=22,000) and evaluated in several test sets using explained variance, mean standardized log loss, skew, and kurtosis. The training dataset excluded subjects with any known psychiatric diagnosis. We transferred the functional normative models to the datasets used in this work for benchmarking (*Table 1*) to generate deviation (Z) scores. HCP was included in the initial training (half of the sample was held out in the test set), while the UMich and COBRE datasets were not included in the training and can be considered as examples of transfer to new, unseen sites.

### "Raw" input data

The data that we compare the output of normative modeling to, referred to throughout this work as 'raw' input data, is simply the outputs of traditional preprocessing methods for structural and functional MRI. For structural MRI, this corresponds to the cortical thickness files that are output after running the Freesurfer recon-all pipeline. We used the aparcstats2table and asegstats2table functions to extract the cortical thickness and subcortical volume from each region in the Destrieux atlas and Freesurfer subcortical atlas. For functional MRI, tradition data refers to the Yeo17 brain network connectomes which were extracted from the normalized, smoothed, de-noised functional time-series. The upper triangle of each subject's symmetric connectivity matrix was vectorized, where each cell represents a unique between-network connection. For clarification, we also note that the raw input data is the starting point of the normative modeling analysis, or in other words, the raw input data is the response variable or independent (Y) variable that is predicted from the vector of covariates when estimating the normative model. Before entering into the benchmarking tasks, to create a fair comparison between raw data and deviation scores, nuisance variables including sex, site, linear and quadratic effects of age and head motion (only for functional models) were regressed out of the raw data (structural and functional) using least squares regression.

### Benchmarking

The benchmarking was performed in three separate tasks, mass univariate group difference testing, multivariate prediction – classification, and multivariate prediction – regression, described in further detail below. In each benchmarking task, a model was estimated using the deviation scores as input features and then estimated again using the raw data as the input features. For task one, group difference testing, the models fit in a univariate approach meaning there was one test performed for each brain feature, and for tasks 2 and 3, classification and regression, the models fit in a multivariate approach. After each model was fit, the performance metrics were evaluated and the difference in performance between the deviation score and raw data models was calculated, again described in more detail below.

### Task one: Mass univariate group difference testing

Mass univariate group difference (schizophrenia versus control) testing was performed across all brain regions. Two sample independent t-tests were estimated and run on the data using the SciPy python package (*Virtanen et al., 2020*). After addressing multiple comparison corrections, brain regions with

FDR corrected p<.05 were considered significant and the total number of regions displaying statistically significant group differences was counted.

For the purpose of comparing group difference effects to individual differences, we also summarized the individual deviation maps and compare this map to the group difference map. Individual deviation maps were summarized by counting the number of individuals with 'extreme' deviations (Z>2 or Z<–2) at a given brain region or network connectivity pair. This was done separately for positive and negative deviations and for each group and visualized qualitatively (Figure 4B). To quantify the individual difference maps in comparison to group differences, we performed a Mann-Whitney U-test on the count of extreme deviations in each group. The U-test was used because the distribution of count data is skewed (non-Gaussian) which the U-test is designed to account for (*Mann and Whitney, 1947*).

## Task two: Multivariate prediction – classification

Support vector machine is a commonly used algorithm in machine learning studies and performs well in classification settings. A support vector machine constructs a set of hyper-planes in a high dimensional space and optimizes to find the hyper-plane that has the largest distance, or margin, to the nearest training data points of any class. A larger margin represents better linear separation between classes and will correspond to a lower error of the classifier in new samples. Samples that lie on the margin boundaries are also called 'support vectors.' The decision function provides per-class scores than can be turned into probabilities estimates of class membership. We used Support vector classification (SVC) with a linear kernel as implemented in the scikit-learn package (version 1.0.9) (*Pedregosa et al., 2011*) to classify a schizophrenia group from a control group. These default hyperparameters were chosen based on following an example of SVC provided by scikit-learn, however, similar results were obtained using a radial basis function kernel (not shown). This classification setting of distinguishing schizophrenia from a control group was chosen due to past work showing the presence of both case-control group differences and individual differences (*Wolfers et al., 2018*). The evaluation metric for the classification task is an area under the receiving operator curve (AUC) averaged across all folds within a 10-fold cross-validation framework.

## Task three: Multivariate prediction – regression

A linear regression model was implemented to predict a latent variable of cognition (general cognitive ability) in the HCP dataset. Brain Basis Set (BBS) is a predictive modeling approach developed and validated in previous studies (*Sripada et al., 2019*; *Sripada et al., 2019*); see also studies by Wager and colleagues for a broadly similar approach (*Chang et al., 2015*; *Wager et al., 2013*; *Woo et al., 2017*). BBS is similar to principal component regression (*Jolliffe, 1982*; *Park, 1981*), with an added predictive element. In the training set, PCA is performed on a $n\_subjects$ by $p\_brain\_features$ matrix using the PCA function from scikit-learn in Python, yielding components ordered by descending eigenvalues. Expression scores are then calculated for each of the $k$ components for each subject by projecting each subject's feature matrix onto each component. A linear regression model is then fit with these expression scores as predictors and the phenotype of interest (general cognitive ability) as the outcome, saving **B**, the $k \times 1$ vector of fitted coefficients, for later use. In a test partition, the expression scores for each of the $k$ components for each subject are again calculated. The predicted phenotype for each test subject is the dot product of **B** learned from the training partition with the vector of component expression scores for that subject. We set k=15 in all models, following prior work (*Rutherford et al., 2020*). The evaluation metric for the regression task is the mean squared error of the prediction in the test set.

## Benchmarking: Model comparison evaluation

Evaluation metrics of each task (count, AUC, and MSE) were calculated independently for both deviation score (Z) and raw data (R) models. Higher AUC, higher count, and lower MSE represent better

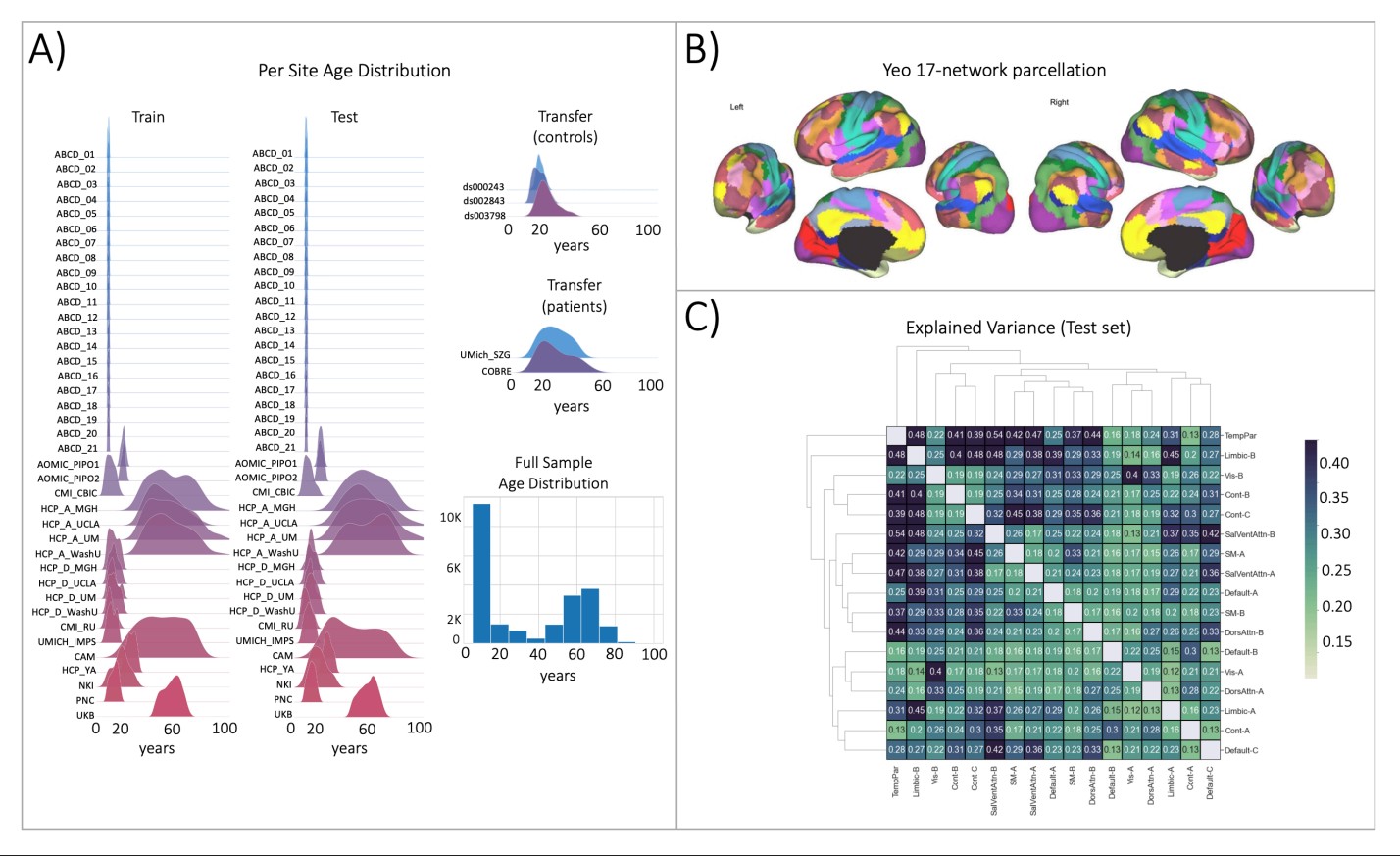

**Figure 2.** Functional brain network normative modeling. (**A**) Age distribution per scanning site in the train, test, and transfer data partitions and across the full sample (train +test). (**B**) The Yeo-17 brain network atlas is used to generate connectomes. Between network connectivity was calculated for all 17 networks, resulting in 136 unique network pairs that were each individually input into a functional normative model. (**C**) The explained variance in the controls test set (N=7244) of each of the unique 136 network pairs of the Yeo-17 atlas. Networks were clustered for visualization to show similar variance patterns.

model performance. We then have a statistic of interest that is observed, theta, which represents the difference between deviation and raw data model performance.

$$\theta_{task\ 1} = Count_z - Count_R$$
$$\theta_{task\ 2} = AUC_z - AUC_R$$
$$\theta_{task\ 3} = MSE_R - MSE_z$$

To assess whether $\theta$ is more likely than would be expected by chance, we generated the null distribution for theta using permutations. Within one iteration of the permutation framework, a random sample is generated by shuffling the labels (In tasks 1 & 2 we shuffle SZ/HC labels, and in task three we shuffle cognition labels). Then this sample is used to train both deviation and raw models, ensuring the same row shuffling scheme across both deviation score and raw data datasets (for each permutation iteration). The shuffled models are evaluated, and we calculate $\theta_{perm}$ for each random shuffle of labels. We set n_permutations =10,000 and use the distribution of $\theta_{perm}$ to calculate a p-value for $\theta_{observed}$ at each benchmarking task. The permuted p-value is equal to (C + 1)/(n_permutations + 1). Where C is the number of permutations where $\theta_{perm} >= \theta_{observed}$. The same evaluation procedure described here 293 (including permutations) was performed for both cortical thickness and functional network modalities.

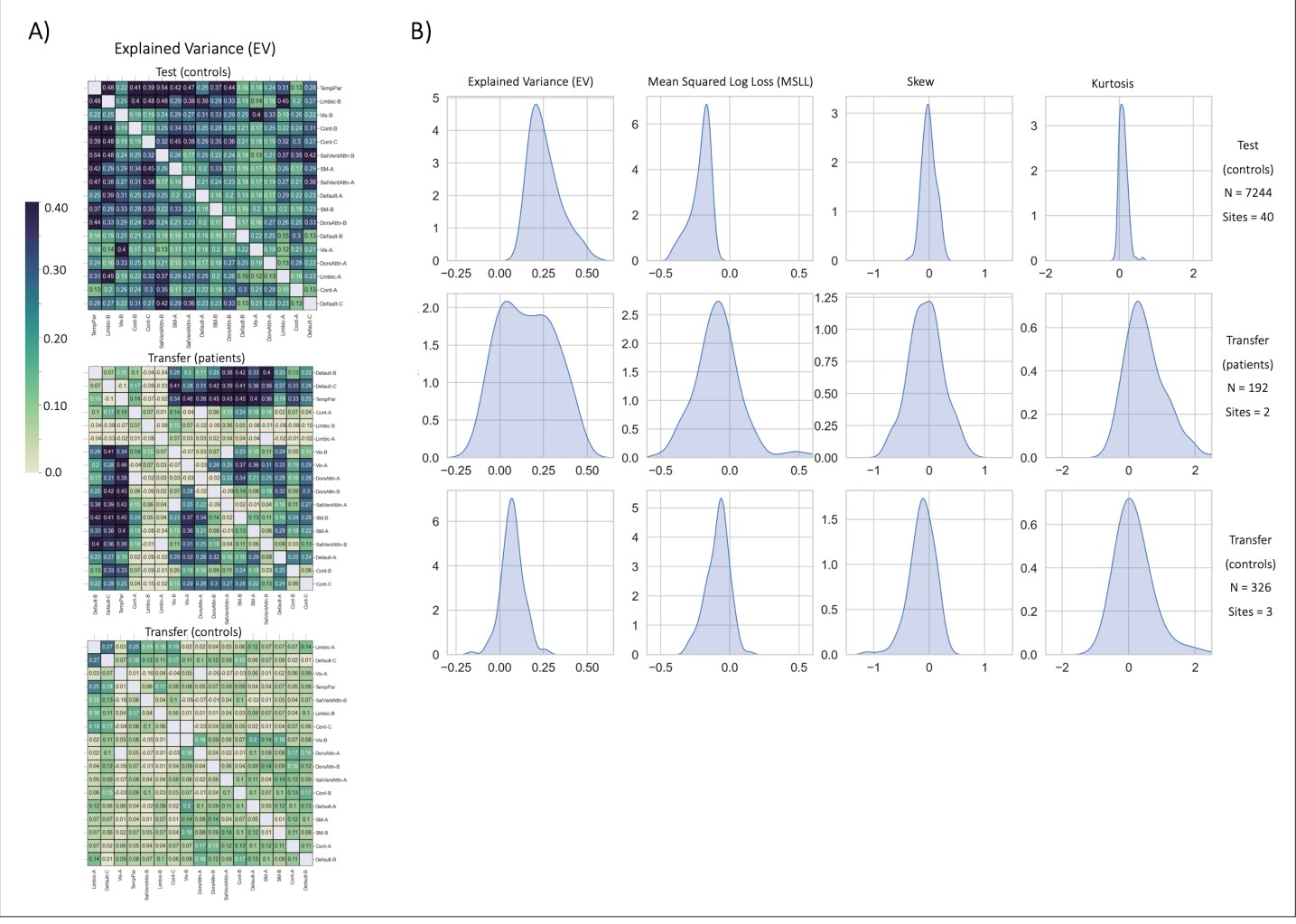

**Figure 3.** Functional normative model evaluation metrics. (**A**) Explained variance per network pair across the test set (top), and both transfer sets (patients – middle, controls – bottom). Networks were clustered for visualization to show similar variance patterns. (**B**) The distribution across all models of the evaluation metrics (columns) in the test set (top row) and both transfer sets (middle and bottom rows). Higher explained variance (closer to one), more negative MSLL, and normally distributed skew and kurtosis correspond to better model fit.

## Results

### Sharing of functional big data normative models

The first result of this work is the evaluation of the functional big data normative models (*Figure 2*). These models build upon the work of *Rutherford et al., 2022a* in which we shared population-level structural normative models charting cortical thickness and subcortical volume across the human lifespan (ages 2–100). The datasets used for training the functional models, the age range of the sample, and the procedures for evaluation closely resemble the structural normative models. The sample size (approx. N=22,000) used for training and testing the functional models is smaller than the structural models (approx. N=58,000) due to data availability (i.e. some sites included in the structural models did not collect functional data or could not share the data) and the quality control procedures (see methods). However, despite the smaller sample size of the functional data reference cohort, the ranges of the evaluation metrics are quite similar to the structural models (*Figure 3*). Most importantly, we demonstrate the opportunity to transfer the functional models to new samples, or sites that were not included in the original training and testing sets, referred to as the transfer set, and show that transfer works well in a clinical sample (*Figure 3* - transfer patients) or sample of healthy controls (*Figure 3* - transfer controls).

**Table 2.** Benchmarking results.

Deviation (Z) score column shows the performance using deviation scores (AUC for classification, the total number of regions with significant group differences FDR-corrected p<0.05 for case versus control, mean squared error for regression), Raw column represents the performance when using the raw data, and Difference column shows the difference between the deviation scores and raw data (Deviation - Raw). Higher AUC, higher count, and lower MSE represent better performance. Positive values in the Difference column show that there is better performance when using deviation scores as input features for classification and group difference tasks, and negative performance difference values for the regression task show there is a better performance using the deviation scores. *=statistically significant difference between Z and Raw established using permutation testing (10 k perms).

| Benchmark | Modality | Normative Modeling Deviation Score Data | Raw Data | Performance Difference |
|---|---|---|---|---|
| Group Difference | Cortical thickness | 117/187 | 0/187 | 117* |
| Group Difference | Functional Networks | 50/136 | 0/136 | 50* |
| Classification | Cortical thickness | 0.87 | 0.43 | 0.44* |
| Classification | Functional Networks | 0.69 | 0.68 | 0.01 |
| Regression | Cortical thickness | 0.699 | 0.708 | −0.008 |
| Regression | Functional Networks | 0.877 | 0.890 | −0.013 |

## Normative modeling shows larger effect sizes in mass univariate group differences

The strongest evidence for embracing normative modeling can be seen in the benchmarking task one group difference (schizophrenia versus controls) testing results (*Table 2*, *Figure 4*). In this application, we observe numerous group differences in both functional and structural deviation score models after applying stringent multiple comparison corrections (FDR p-value <0.05). The strongest effects (HC>SZ) in the structural models were located in the right hemisphere lateral occipitotemporal sulcus (S_oc_temp_lat) thickness, right hemisphere superior segment of the circular sulcus of the insula (S_circular_ins_sup) thickness, right Accumbens volume, left hemisphere Supramarginal gyrus (G_pariet_inf_Supramar) thickness, and left hemisphere Inferior occipital gyrus (O3) and sulcus (G_and_S_occipital_inf) thickness. For the functional models, the strongest effects (HC>SZ t-statistic) were observed in the between-network connectivity of Visual A-Default B, Dorsal Attention A-Control B, and Visual B-Limbic A. In the raw data models, which were residualized of covariates including site, sex, and linear +quadratic effects of age and head motion (only included for functional models), we observe no group differences after multiple comparison corrections. The lack of any group differences in the raw data was initially a puzzling finding due to reported group differences in the literature (*Arbabshirani et al., 2013*; *Cetin et al., 2015*; *Cetin et al., 2016*; *Cheon et al., 2022*; *Dansereau et al., 2017*; *Howes et al., 2023*; *Lei et al., 2020a*; *Lei et al., 2020b*; *Meng et al., 2017*; *Rahim et al., 2017*; *Rosa et al., 2015*; *Salvador et al., 2017*; *Shi et al., 2021*; *van Erp et al., 2018*; *Venkataraman et al., 2012*; *Wannan et al., 2019*; *Yu et al., 2012*), however, upon the investigation of the uncorrected statistical maps, we observe that the raw data follows a similar pattern to the deviation group difference map (*Figure 4*), but these results do not withstand multiple comparison correction. For full statistics including the corrected and uncorrected p-values and test-statistic of every ROI, see *Supplementary files 3 and 4*. While there have been reported group differences between controls and schizophrenia in cortical thickness and resting state brain networks in the literature, these studies have used different datasets (of varying sample sizes), different preprocessing pipelines and software versions, and different statistical frameworks (*Castro et al., 2016*; *Di Biase et al., 2019*; *Dwyer et al., 2018*; *Geisler et al., 2015*; *Marek et al., 2022*; *Sui et al., 2015*; *van Haren et al., 2011*). When reviewing the literature of studies on SZ versus HC group difference testing, we did not find any study that performed univariate t-testing and multiple comparison correction at the ROI-level or network-level, rather most works used statistical tests and multiple comparison correction at the voxel-level or edge-level. Combined with the known patterns of heterogeneity present in schizophrenia disorder (*Lv et al., 2021*; *Wolfers et al., 2018*), it is unsurprising that our results differ from past studies.

The qualitative (*Figure 4B*) and quantitative (*Figure 4C*) comparison of the group difference maps with the individual difference maps showed the additional benefit of normative modeling - that it can

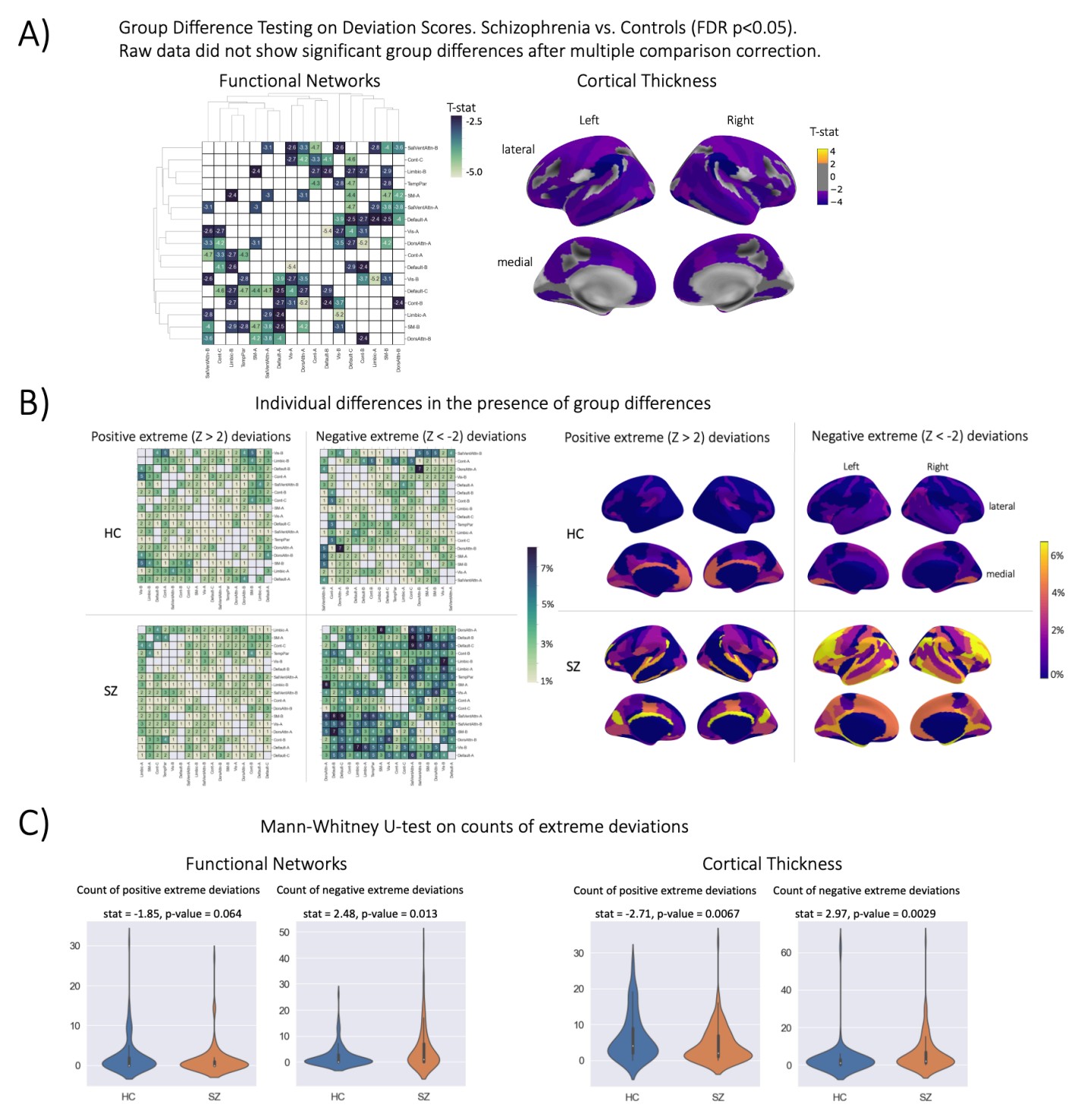

**Figure 4.** Group difference testing evaluation. (**A**) Significant group differences in the deviation score models, (top left) functional brain network deviation, and (top right) cortical thickness deviation scores. The raw data, either cortical thickness or functional brain networks (residualized of sex and linear/ quadratic effects of age and motion (mean framewise displacement)) resulted in no significant group differences after multiple comparison corrections. Functional networks were clustered for visualization to show similar variance patterns. (**B**) There are still individual differences observed that do not overlap with the group difference map, showing the benefit of normative modeling, which can detect both group and individual differences through proper modeling of variation. Functional networks were clustered for visualization to show similar variance patterns. (**C**) There are significant group differences in the summaries (count) of the individual difference maps (panel B).

reveal subtle individual differences which are lost when only looking at group means. The individual difference maps show that at every brain region or connection, there is at least one person, across both patient and clinical groups, that has an extreme deviation. We found significant differences in the count of negative deviations (SZ >HC) for both cortical thickness (p=0.0029) and functional networks (p=0.013), and significant differences (HC >SZ) in the count of positive cortical thickness (p=0.0067).

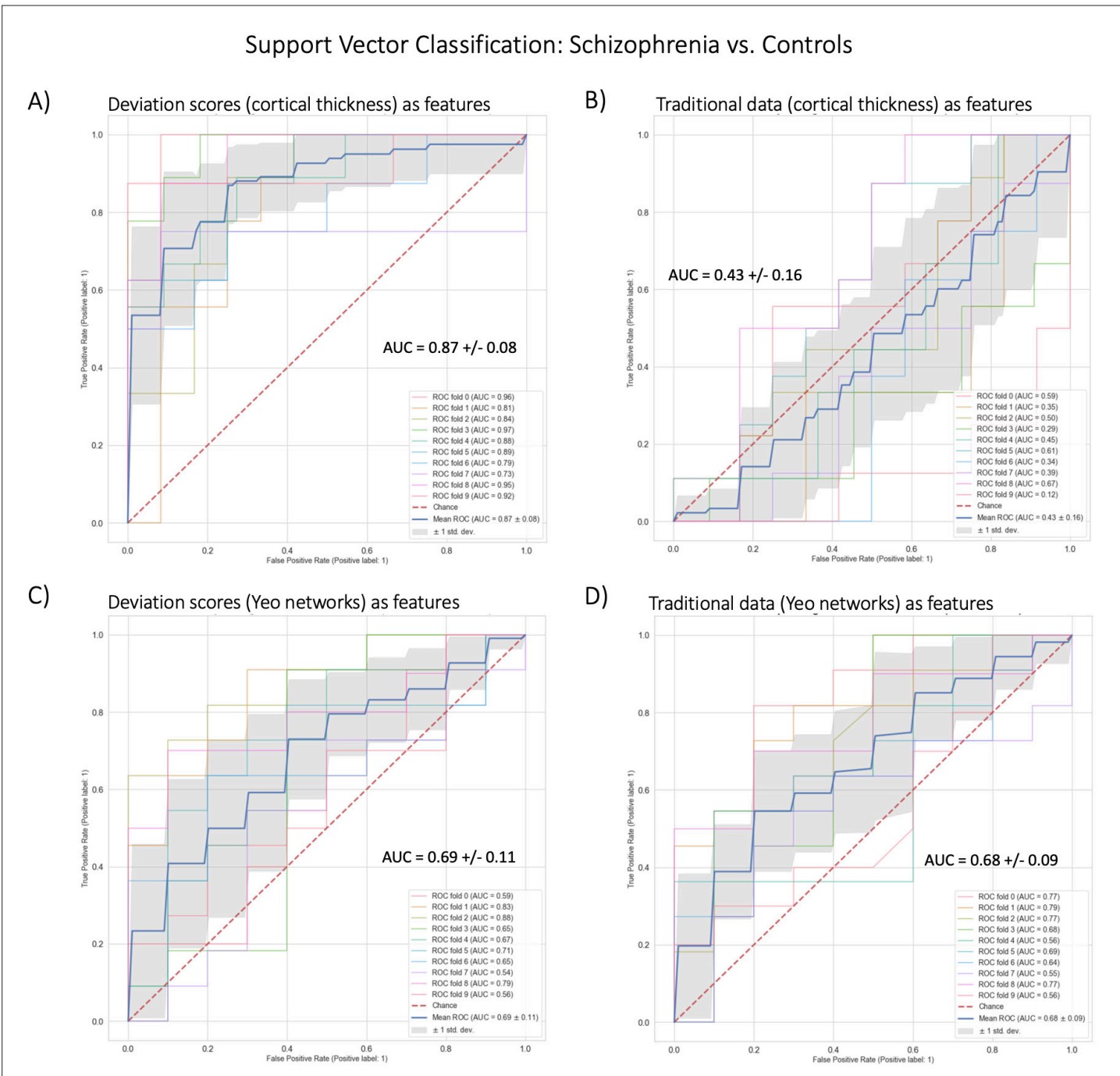

**Figure 5.** Benchmark task two multivariate prediction – Classification evaluation. (**A**) Support vector classification (SVC) using cortical thickness deviation scores as input features (most accurate model). (**B**) SVC using cortical thickness (residualized of sex and linear/quadratic effects of age) as input features. (**C**) SVC using functional brain network deviation scores as input features. (**D**) SVC using functional brain networks (residualized of sex and linear/quadratic effects of age and motion (mean framewise displacement)) as input features.

## Normative modeling shows highest classification performance using cortical thickness

In benchmarking task two, we classified schizophrenia versus controls using SVC within a 10-fold cross-validation framework (*Table 2*, *Figure 5*). The best-performing model used cortical thickness deviation scores to achieve a classification accuracy of 87% (AUC = 0.87). The raw cortical thickness model accuracy was indistinguishable from chance accuracy (AUC = 0.43). The AUC performance difference between the cortical thickness deviation and raw data models was 0.44, and this performance difference was statistically significant. The functional models, both deviation scores (0.69) and raw data (0.68) were more accurate than chance accuracy, however, the performance difference (i.e. improvement in accuracy using the deviation scores) was small (0.01) and was not statistically significant.

## Normative modeling shows modest performance improvement in predicting cognition

In benchmarking task three we fit multivariate predictive models in a held-out test set of healthy individuals in the Human Connectome Project young-adult study to predict general cognitive ability (*Table 2*). The evidence provided by this task weakly favors the deviation score models. The most accurate (lowest mean squared error) model was the deviation cortical thickness model (MSE = 0.699). However, there was only an improvement of 0.008 in the deviation score model compared to the raw data model (MSE = 0.708) and this difference was not statistically significant. For the functional models, both the deviation score (MSE = 0.877) and raw data (MSE = 0.890) models were less accurate than the structural models and the difference between them (0.013) was also not statistically significant.

## Discussion

This work expands the available open-source tools for conducting normative modeling analyses and provides clear evidence for why normative modeling should be utilized by the neuroimaging community (and beyond). We updated our publicly available repository of pre-trained normative models to include a new MRI imaging modality (models of resting-state functional connectivity extracted from the Yeo-17 and Smith-10 brain network atlases) and demonstrate how to transfer these models to new data sources. The repository includes an example transfer dataset and in addition, we have developed a user-friendly interface (https://pcnportal.dccn.nl/) that allows transferring the pre-trained normative models to new samples without requiring any programming. Next, we compared the features that are output from normative modeling (deviation scores) against 'raw' data features across several benchmarking tasks including univariate group difference testing (schizophrenia vs. control), multivariate prediction – classification (schizophrenia vs. control), and multivariate prediction – regression (predicting general cognitive ability). We found across all benchmarking tasks there were minor (regression) to strong (group difference testing) benefits of using deviation scores compared to the raw data features.

The fact that the deviation score models perform better than the raw data models confirms the utility of placing individuals into reference models. Our results show that normative modeling can capture population trends, uncover clinical group differences, and preserve the ability to study individual differences. We have some intuition on why the deviation score models perform better on the benchmarking tasks than the raw data. With normative modeling, we are accounting for many sources of variance that are not necessarily clinically meaningful (i.e. site) and we are able to capture clinically meaningful information within the reference cohort perspective. The reference model helps beyond just removing confounding variables such as scanner noise because we show that even when removing the nuisance covariates (age, sex, site, head motion) from the raw data, the normative modeling features still perform better on the benchmarking tasks.

Prior works on the methodological innovation and application of normative modeling *Kia et al., 2018*; *Kia et al., 2020*; *Kia et al., 2021*; *Kia and Marquand, 2018* have focused on the beginning foundational steps of the framework (i.e. data selection and preparation, algorithmic implementation, and carefully evaluating out of sample model performance). However, the framework does not end after the model has been fit to the data (estimation step) and performance metrics have been established (evaluation step). Transferring the models to new samples, interpretation of the results,

and potential downstream analysis are equally important steps, but they have received less attention. When it comes time to interpret the model outputs, it is easy to fall back into the case-control thinking paradigm, even after fitting a normative model to one's data (which is supposed to be an alternative to case vs. control approaches). This is due in part to the challenges arising from the results existing in a very high dimensional space (~100 s to 1000 s of brain regions from ~100 s to 1000 s of subjects). There is a reasonable need to distill and summarize these high-dimensional results. However, it is important to remember there is always a trade-off between having a complex enough of a model to explain the data and dimensionality reduction for the sake of interpretation simplicity. This distillation process often leads back to placing individuals into groups (i.e. case-control thinking) and interpreting group patterns or looking for group effects, rather than interpreting results at the level of the individual. We acknowledge the value and complementary nature of understanding individual variation relative to group means (case-control thinking) and clarify that we do not claim the superiority of normative modeling over case-control methods. Rather, our results from this work, especially in the comparisons of group difference map to individual difference maps (*Figure 4*), show that the outputs of normative modeling can be used to validate, refine, and further understand some of the inconsistencies in previous findings from case-control literature.

There are several limitations of the present work. First, the representation of functional normative models may be surprising and concerning. Typically, resting-state connectivity matrices are calculated using parcellations containing between 100–1000 nodes and 5000–500,000 connections. However, the Yeo-17 atlas (*Yeo et al., 2011*) was specifically chosen because of its widespread use and the fact that many other (higher resolution) functional brain parcellations have been mapped to the Yeo brain networks (*Eickhoff et al., 2018*; *Glasser et al., 2016*; *Gordon et al., 2016*; *Gordon et al., 2017*; *Kong et al., 2019*; *Laumann et al., 2015*; *Power et al., 2011*; *Schaefer et al., 2018*; *Shen et al., 2013*). There is an on-going debate about the best representation of functional brain activity. Using the Yeo-17 brain networks to model functional connectivity ignores important considerations regarding brain dynamics, flexible node configurations, overlapping functional modes, hard versus soft parcellations, and many other important issues. We have also shared functional normative models using the Smith-10 ICA-based parcellation, though did not repeat the benchmarking tasks using these data. Apart from our choice of parcellation, there are fundamental open questions regarding the nature of the brain's functional architecture, including how it is defined and measured. While it is outside the scope of this work to engage in these debates, we acknowledge their importance and refer curious readers to a thorough review of functional connectivity challenges (*Bijsterbosch et al., 2020*).

We would also like to expand on our prior discussion (*Rutherford et al., 2022a*) on the limitations of the reference cohort demographics, and the use of the word 'normative.' The included sample for training the functional normative models in this work, and the structural normative modeling sample in *Rutherford et al., 2022a* are most likely overrepresentative of European-ancestry (WEIRD population *Henrich et al., 2010*) due to the data coming from academic research studies, which do not match global population demographics. Our models do not include race or ethnicity as covariates due to data availability (many sites did not provide race or ethnicity information). Prior research supports the use of age-specific templates and ethnicity-specific growth charts (*Dong et al., 2020*). This is a major limitation that requires additional future work and should be considered carefully when transferring the model to diverse data (*Benkarim et al., 2022*; *Greene et al., 2022*; *Li et al., 2022*). The term 'normative model' can be defined in other fields in a very different manner than ours (*Baron, 2004*; *Colyvan, 2013*; *Titelbaum, 2021*). We clarify that ours is strictly a statistical notion (normative = being within the central tendency for a population). Critically, we do not use normative in a moral or ethical sense, and we are not suggesting that individuals with high deviation scores require action or intervention to be pulled toward the population average. Although in some cases this may be true, we in no way assume that high deviations are problematic or unhealthy (they may in fact represent compensatory changes that are adaptive). In any case, we treat large deviations from statistical normality strictly as markers predictive of clinical states or conditions of interest.

There are many open research questions regarding normative modeling. Future research directions are likely to include: (1) further expansion of open-source pre-trained normative modeling repositories to include additional MRI imaging modalities such as task-based functional MRI and diffusion-weighted imaging, other neuroimaging modalities such as EEG or MEG, and models that include

other non-biological measures, (2) increase in the resolution of existing models (i.e. voxel, vertex, models of brain structure and higher resolution functional parcellations), (3) replication and refinement of the proposed benchmarking tasks in other datasets including hyperparameter tuning and different algorithm implementation, and improving the regression benchmarking task, and (4) including additional benchmarking tasks beyond the ones considered here.

There has been recent interesting work on 'failure analysis' of brain-behavior models (*Greene et al., 2022*), and we would like to highlight that normative modeling is an ideal method for conducting this type of analysis. Through normative modeling, research questions such as 'what are the common patterns in the subjects that are classified well versus those that are not classified well' can be explored. Additional recent work (*Marek et al., 2022*) has highlighted important issues the brain-behavior modeling community must face, such as poor reliability of the imaging data, poor stability and accuracy of the predictive models, and the very large sample sizes (exceeding that of even the largest neuroimaging samples) required for accurate predictions. There has also been working showing that brain-behavior predictions are more reliable than the underlying functional data (*Taxali et al., 2021*), and other ideas for improving brain-behavior predictive models are discussed in-depth here (*Finn and Rosenberg, 2021*; *Rosenberg and Finn, 2022*). Nevertheless, we acknowledge these challenges and believe that sharing pre-trained machine learning models and further development of transfer learning of these models could help further address these issues.

In this work, we have focused on the downstream steps of the normative modeling framework involving evaluation and interpretation, and how insights can be made on multiple levels. Through the precise modeling of different sources of variation, there is much knowledge to be gained at the level of populations, clinical groups, and individuals.

## Additional information

### Competing interests

Christian F Beckmann: is director and shareholder of SBGNeuro Ltd. Henricus G Ruhe: received speaker's honorarium from Lundbeck and Janssen. The other authors declare that no competing interests exist.

### Funding

| Funder | Grant reference number | Author |
|---|---|---|
| European Research Council | 10100118 | Andre F Marquand |
| Wellcome Trust | 215698/Z/19/Z | Andre F Marquand |
| Wellcome Trust | 098369/Z/12/Z | Christian F Beckmann |
| National Institute of Mental Health | R01MH122491 | Ivy F Tso |
| National Institute of Mental Health | R01MH123458 | Chandra Sripada |
| National Institute of Mental Health | R01MH130348 | Chandra Sripada |

The funders had no role in study design, data collection and interpretation, or the decision to submit the work for publication. For the purpose of Open Access, the authors have applied a CC BY public copyright license to any Author Accepted Manuscript version arising from this submission.

### Author contributions

Saige Rutherford, Conceptualization, Resources, Data curation, Software, Formal analysis, Validation, Methodology, Writing - original draft, Writing – review and editing; Pieter Barkema, Resources, Software, Writing – review and editing; Ivy F Tso, Chandra Sripada, Christian F Beckmann, Henricus G Ruhe, Andre F Marquand, Supervision, Funding acquisition, Writing – review and editing

### Author ORCIDs
Saige Rutherford http://orcid.org/0000-0003-3006-9044
Chandra Sripada http://orcid.org/0000-0001-9025-6453

### Ethics
Human subjects: Secondary data analysis was conducted in this work. Data were pooled from multiple data sources described in the supplemental tables. All subjects provided informed consent. Subject recruitment procedures and informed consent forms, including consent to share de-identified data, were approved by the corresponding university institutional review board where data were collected. Human subjects: Ethical approval for the public data were provided by the relevant local research authorities for the studies contributing data. For full details, see the main study publications in the main text. For all clinical studies, approval was obtained via the local ethical review authorities, i.e., Delta: The local ethics committee of the Academic Medical Center of the University of Amsterdam (AMC-METC) Nr.:11/050, UMich_IMPS: University of Michigan Institution Review Board HUM00088188, UMich_SZG: University of Michigan Institution Review Board HUM00080457.

### Decision letter and Author response
Decision letter https://doi.org/10.7554/eLife.85082.sa1
Author response https://doi.org/10.7554/eLife.85082.sa2

## Additional files

### Supplementary files
• Supplementary file 1. Functional Normative Model Demographics. *Description:* For each included site, we show the sample size (N), age (mean, standard deviation), and sex distribution (Female/Male percent) in the training set (shown in blue) and testing set (shown in green) of the normative models of functional connectivity between large scale resting-state brain networks from the Yeo 17 network atlas.

• Supplementary file 2. Surface Area Normative Model Demographics. *Description:* For each included site, we show the sample size (N), age (mean, standard deviation), and sex distribution (Female/Male percent) of the normative models of surface area extracted for all regions of interest in the Destrieux Freesurfer atlas.

• Supplementary file 3. Structural Group Difference Testing Statistics. *Description:* We show for all cortical thickness and subcortical volume from the Destrieux and aseg Freesurfer atlases regions of interest (ROIs from a two-sample t-test between Schizophrenia versus Healthy Controls) the t-statistic (T-stat), False Discovery Rate corrected p-value (FDRcorr_pvalue), and uncorrected p-value (uncorr_pvalue) for both the raw data (shown in green) and the deviation scores (shown in blue).

• Supplementary file 4. Functional Connectivity Group Difference Testing Statistics. *Description:* We show for all Yeo-17 between network connectivity regions of interest (ROIs) from a two-sample t-test between Schizophrenia versus Healthy Controls the t-statistic (T-stat), False Discovery Rate corrected p-value (FDRcorr_pvalue), and uncorrected p-value (uncorr_pvalue) for both the raw data (shown in green) and the deviation scores (shown in blue).

• MDAR checklist

### Data availability
Pre-trained normative models are available on GitHub (https://github.com/predictive-clinical-neuroscience/braincharts, (copy archived at swh:1:rev:299e126ff053e2353091831a888c3ccd1ca6edeb)) and Google Colab (https://colab.research.google.com/github/predictive-clinical-neuroscience/braincharts/blob/master/scripts/apply_normative_models_yeo17.ipynb). Scripts for running the benchmarking analysis and visualizations are available on GitHub (https://github.com/saigerutherford/evidence_embracing_nm, (copy archived at swh:1:rev:1b4198389e2940dd3d10055164d68d46e0a20750)). An online portal for running models without code is available (https://pcnportal.dccn.nl).

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

## Appendix 1

### Functional MRI Acquisition Parameters

In the HCP study, four runs of resting state fMRI data (14.5 min each) were acquired on a Siemens Skyra 3 Tesla scanner using multi-band gradient-echo EPI (TR = 720ms, TE = 33ms, flip angle = 52°, multiband acceleration factor = 8, 2 mm isotropic voxels, FOV = 208 × 180 mm, 72 slices, alternating RL/LR phase encode direction). T1 weighted scans were acquired with 3D MPRAGE sequence (TR = 2400ms, TE = 2.14ms, TI = 1000ms, flip angle = 8, 0.7 mm isotropic voxels, FOV = 224 mm, 256 sagittal slices) and T2 weighted scans were acquired with a SPACE sequence (TR = 3200ms, TE = 565ms, 0.7 mm isotropic voxels, FOV = 224 mm, 256 sagittal slices). In the COBRE study, the T1 weighted acquisition is a multi-echo MPRAGE (MEMPR) sequence (1 mm isotropic). Resting state functional MRI data was collected with single-shot full k-space echo-planar imaging (EPI) (TR = 2000ms, TE = 29ms, FOV = 64 × 64, 32 slices in axial plane interleaved multi slice series ascending, voxel size = 3 × 3 x4 mm$^3$). The University of Michigan SchizGaze study was collected in two phases with different parameters but using the same MRI machine (3.0T GE MR 750 Discovery scanner). In SchizGaze1 (N=47), functional images were acquired with a T2*-weighted, reverse spiral acquisition sequence (TR = 2000ms, 240 volumes (8 min), 3 mm isotropic voxels) and a T1-weighted image was acquired in the same prescription as the functional images to facilitate co-registration. In SchizGaze2 (N=68), functional images were acquired with a T2*-weighted multi-band EPI sequence (multi-band acceleration factor of 8, TR = 800ms, 453 volumes (6 min), 2.4 mm isotropic voxels) and T1w (MPRAGE) and T2w structural scans were acquired for co-registration with the functional data. In addition, field maps were acquired to correct for intensity and geometric distortions.

### Functional MRI Preprocessing Methods

T1w images are corrected for intensity nonuniformity, reconstructed with recon-all (FreeSurfer), spatially normalized (ANTs), and segmented with FAST (FSL). For every BOLD run, data are co-registered to the corresponding T1w reference, and the BOLD signal is sampled onto the subject's surfaces with mri_vol2surf (FreeSurfer). A set of noise regressors are generated during the preceding steps that are used to remove a number of artifactual signals from the data during subsequent processing, and these noise regressors include: head-motion parameters (via MCFLIFT; FSL) framewise displacement and DVARS, and physiological noise regressors for use in component-based noise correction (CompCor). ICA-based denoising is implemented via ICA-AROMA and we compute 'non-aggressive' noise regressors. Resting state connectomes are generated from the fMRIPrep processed resting state data using Nilearn, denoising using the noise regressors generated above, with orthogonalization of regressors to avoid reintroducing artifactual signals.

### Functional Brain Networks Normative Modeling

Data from 40 sites were combined to create the initial full sample. These sites are described in detail in **Supplementary file 1**, including the sample size, age (mean and standard deviation), and sex distribution of each site. Many sites were pulled from publicly available datasets including ABCD, CAMCAN, CMI-HBN, HCP-Aging, HCP-Development, HCP-Early Psychosis, HCP-Young Adult, NKI-RS, OpenNeuro, PNC, and UKBiobank. For datasets that include repeated visits (i.e. ABCD, UKBiobank), only the first visit was included. Full details regarding sample characteristics, diagnostic procedures and acquisition protocols can be found in the publications associated with each of the studies. Training and testing datasets (80/20) were created using scikit-learn's train_test_split function, stratifying on the site variable. To show generalizability of the models to new data not included in training, we leveraged three datasets (ds000243, ds002843, ds003798) from OpenNeuro.org to create a multi-site transfer dataset.

Normative modeling was run using python 3.8 and the PCNtoolkit package (version 0.26). Bayesian Linear Regression (BLR) with likelihood warping was used to predict each Yeo-17 network pair from a vector of covariates (age, sex, mean_FD, site). For a detailed mathematical description see **Fraza et al., 2021**. Briefly, for each region of interest, $y$ is predicted as:

$$y = w^T \phi(x) + \epsilon \tag{1}$$

Where $w^T$ is the estimated weight vector, $\phi(x)$ is a basis expansion of the of covariate vector **x,** consisting of a B-spline basis expansion (cubic spline with 5 evenly spaced knots) to model non-linear effects of age, and $\epsilon = \eta(0, \beta)$ a Gaussian noise distribution with mean zero and noise precision term β (the inverse variance). A likelihood warping approach was used to model non-Gaussian effects by

applying a bijective nonlinear warping function to the non-Gaussian response variables to map them to a Gaussian latent space where inference can be performed in closed form. We used a 'sinarcsinsh' warping function, equivalent to the SHASH distribution that is commonly used in the generalized additive modeling literature (*Jones and Pewsey, 2009*). Site variation was modeled using fixed effects. A fast numerical optimization algorithm was used to optimize hyperparameters ('Powell'). Deviation scores (Z-scores) are calculated for the *n-th* subject, and *d-th* brain area, in the test set as:

$$Z_{nd} = \frac{y_{nd} - \hat{y}_{nd}}{\sqrt{\sigma_d^2 + (\sigma_*^2)_d}}$$
(2)

Where $y_{nd}$ is the true response, $\hat{y}_{nd}$ is the predicted mean, $\sigma_d^2$ is the estimated noise variance (reflecting uncertainty in the data), and $(\sigma_*^2)_d$ is the variance attributed to modeling uncertainty. Model fit for each brain region was evaluated by calculating the explained variance (which measures central tendency), the mean squared log-loss (MSLL, central tendency and variance) plus skew and kurtosis of the deviation scores (2) which measures how well the shape of the regression function matches the data (*Dinga et al., 2021*).

All pretrained models and code are shared online including documentation for transferring to new sites and an example transfer dataset. Given a new set of data (e.g. sites not present in the training set), this is done by first applying the warp parameters estimating on the training data to the new dataset, adjusting the mean and variance in the latent Gaussian space, then (if necessary) warping the adjusted data back to the original space, which is similar to the approach outlined in *Dinga et al., 2021*.

