## [Editor Report]

This is a rigorous and compelling extension of previous normative modeling work. The current study demonstrates that normative models incorporating lifespan trajectories of structural and functional connectivity provide a strong basis for brain imaging studies across a range of tasks including, univariate group difference assessment, classification, and building regression models. The work is important, rigorous and a valuable contribution to the field.

---

## [Decision Letter]

**Decision letter after peer review:**

Thank you for submitting your article "Evidence for Embracing Normative Modeling" for consideration by *eLife*. Your article has been reviewed by 3 peer reviewers, and the evaluation has been overseen by a Reviewing Editor and Chris Baker as the Senior Editor. The following individuals involved in the review of your submission have agreed to reveal their identity: Todd Constable (Reviewer #1); Oscar Esteban (Reviewer #3).

Essential revisions:

While the reviewers are convinced of the potential importance of this extension of your earlier work, they raise some comments about the clarity of aspects of the. In your revision, it is important for you to consider each of these issues and revise the manuscript as appropriate.

*Reviewer #1 (Recommendations for the authors):*

This is a clearly written and scientifically rigorous presentation of the utility of normative modeling with structural and functional MR data. I have no suggestions for improving the work and find it acceptable for publication in its current form.

*Reviewer #2 (Recommendations for the authors):*

Please clarify: Are all normative models (i.e. across all modalities) based on the same set of covariates?

What exactly is the input to models in the benchmarking tasks? I.e. is it a single value (deviation score) per individual for the normative model but the full functional network for the raw data model?

As mentioned in the manuscript, the null findings for raw data models in the group difference task are surprising. Could you elaborate on the difference between cited work that has found significant effects? (E.g. differences in sample sizes, differences in preprocessing steps, did previous work not include multiple comparisons corrections?)

The choice of a linear kernel for the SVM classification task is not clear to me. Since the input vectors are low dimensional, would a non-linear (e.g. rbf) kernel not perform better?

Is there any expected reason why classification in the case of the raw data models would work on the functional but not on the structural measures? (Generally, I would expect a higher noise level in the functional data and thus a more challenging classification scenario.)

In line 415ff, you mention that normative models allow for the interrogation of patterns in classified vs misclassified samples. Have you seen any such patterns in the schizophrenia vs control benchmark test? (If so, I think it would be instructive for readers to include a brief discussion of observed patterns for the schizophrenia vs controls application.)

*Reviewer #3 (Recommendations for the authors):*

I believe this manuscript is in an advanced maturity status; therefore, my recommendations are not particularly major. In no particular order:

– Colab examples and tutorials: for some reason, the pnctoolkit package cannot be installed due to dependencies issues. Unfortunately, that dissuaded me from executing the examples (which I was really looking forward to doing and, I believe, is a significant aspect of the overall work).

– Abstract. The sentence "Across all benchmarks, we confirm the advantage (i.e., stronger effect sizes, more accurate classification, and prediction) of using normative modeling features" is bordering the overstatement. I would suggest bringing up more specific aspects of the results (more quantitative statements, if you will).

– Introduction (ll. 63--72): this paragraph feels a bit out of place. I think the intent is to connect the last sentence (69--72), which I agree is relevant. However, I am not convinced by the flow of the paragraph (e.g., not sure the references to Gau et al. (2021) and Levitis et al. (2021) are a good fit here, as opposed to a discussion). I would invite the authors to rethink how this paragraph is carried out.

– Table 1. HCP's sex ratios. I believe it should say 46.6% of males.

– Figure 1. I suggest annotating the different tasks within (C) and (D), and updating the caption accordingly.

– Methods (l. 203). For ignorant readers like this reviewer, a reduced mention of why this particular test (Mann-Whitney U-test) was chosen would be appreciated. Perhaps some further details could also be given in the appendix.

– Methods (l. 212). Some elaboration on how the hyperparameters were selected (i.e., kernel and C constant). I have looked into the corresponding jupyter notebook (which was of course, available, thanks!), and it seems the defaults were used. Could this have an impact on the advantages of normative modeling?

– Methods (l. 232) Evaluation. I would suggest (i) moving the metrics within the body of each task previously (i.e.,) the definition of $\theta$ in equations. (1, 2, 3); and (ii) keeping a subsection with ll. 244--254 with perhaps a more detailed section title (e.g., statistical inference --or evaluation-- of model comparisons).

– Results section: I would suggest the authors provide subsection titles that advance the contents of the subsection. E.g., in line 271, something along the lines of "Normative modeling showcased larger effect sizes in mass univariate group differences."

– ll. 290--292. I suggest reflecting this in Figure 4A, showcasing the corresponding results without multiple comparisons correction (which should be clearly indicated) to give further support to the idea that they were along the same lines.

– ll. 285--286. "The lack of any group differences in the raw data was initially a puzzling finding…" Can this be interpreted as normative models being more powered in the discussion? This is sideways mentioned when indicating that normative models have a harmonization effect on the individual metrics, more effective than regressing out obvious confounders. A more frontal/intentional incursion into this discussion would be appreciated.

– l. 301. The subsection hierarchy seems broken. It is confusing that this particular result is not at the same level as the other (e.g., l. 311).

– l. 365. That "from this work" seemed out of place.

---

## [Author Response]

Reviewer #2 (Recommendations for the authors):Please clarify: Are all normative models (i.e. across all modalities) based on the same set of covariates?

The normative models for structure and function were fit using the same demographic variables (age and sex). Both structure and functional models also used MRI scanner site as a covariate, and both included a metric representing data quality (Euler number for structural models and mean framewise displacement for functional models). We have clarified this in the manuscript.

“The covariates used to train the structural normative models included age, sex, data quality metric (Euler number), and site… The covariates used to train the functional normative models were similar as the structural normative models which included age, sex, data quality metric (mean framewise displacement), and site.”

What exactly is the input to models in the benchmarking tasks? I.e. is it a single value (deviation score) per individual for the normative model but the full functional network for the raw data model?

The input into the benchmarking tasks is the same dimensionality for the deviation scores and raw data. For the structural models, each subject has 187 features, which represent all of the ROIs in the Destrieux atlas combined with all of the ROIs in the subcortical Freesurfer atlas. For the functional models, each subject has 136 features, which represent all of the 17 Yeo between brain network connectivity pairs. We have clarified this further in the text.

“For task 1, group difference testing, the models were fit in a univariate approach meaning there was one test performed for each brain feature and for tasks 2 and 3, classification and regression, the models were fit in a multivariate approach.”

As mentioned in the manuscript, the null findings for raw data models in the group difference task are surprising. Could you elaborate on the difference between cited work that has found significant effects? (E.g. differences in sample sizes, differences in preprocessing steps, did previous work not include multiple comparisons corrections?)

We found several papers that used the same COBRE dataset (1-3 described below) and looked at structural differences and classification (4 described below) between SZ and HC groups. While some of these papers report group differences, these papers do not directly perform t-tests on the cortical thickness ROI-level or voxel-level data. They all use much fancier approaches such as clustering, CCA, and non-linear ICA, as well as difference multiple comparison techniques and methods for preprocessing the data. There are no published works showing group differences in cortical thickness in the second dataset we used (UMich). We also further explored publications (using different datasets) that have directly performed ttests on cortical thickness data between SZ and HC groups and found significant group differences (https://www.ncbi.nlm.nih.gov/pmc/articles/PMC6948361/, https://academic.oup.com/schizophreniabulletin/article/45/4/911/5095722, and https://jamanetwork.com/journals/jamapsychiatry/fullarticle/1107282). We noticed these works performed voxel-level tests and used cluster-based multiple comparison correction methods. We could not find any publications that performed the t-tests and multiple comparison correction at the ROI-level, as we did in this work. We also point out work establishing the heterogeneity of cortical thickness in schizophrenia (https://www.nature.com/articles/s41380-020-00882-5), and we consider this to likely be a cause of these varying results in the literature (along with the varying methods used across studies). We have added these references and summarized these ideas in the Discussion section of our paper.

1. https://www.ncbi.nlm.nih.gov/pmc/articles/PMC4705852/#SD1

They use a much older FS version (4.0.1), and a different method for multiple comparison correction (Monte Carlo simulations 10.1016/j.neuroimage.2006.07.036). They took a cluster approach and found significant group differences in 3 out of 4 clusters (reduced cortical thickness in SZ compared to HC). They do not directly do t-tests on the cortical thickness images alone (only on the clusters which are combined with behavioral data).

2. https://www.ncbi.nlm.nih.gov/pmc/articles/PMC4547923/

They use Gray Matter Volume (GMV) (and other imaging modalities) within a CCA framework combined with cognition behavioral data. Data were preprocessed in SPM8, and GMV images were smoothed with an 8mm kernel. All images were also normalized to have the same average sum of squares. They do not directly do t-tests on the GM images alone.

3. https://www.ncbi.nlm.nih.gov/pmc/articles/PMC4965265/

They use non-linear ICA, Variational Autoencoders, and regular ICA to look at structural abnormalities between SZ and HC. They find group differences in 4 out of 30 non-linear ICA components, and 2/20 VAE/ICA components. Structural data were preprocessed using SPM (unspecified version) and were smoothed with an 8mm Gaussian kernel.

4. https://www.ncbi.nlm.nih.gov/pmc/articles/PMC6101481/

They also merged the COBRE dataset with another SZ dataset collected in Germany. They do not perform group difference testing but use a linear SVM to classify SZ vs. HC (68.3% accuracy) and performed further sub-group analysis based on symptom scores and other behavioral variables. Gray Matter Volume images were used and were preprocessed using SPM and DARTEL.

“While there have been reported group differences between controls and schizophrenia in cortical thickness and resting state brain networks in the literature, these studies have used different data sets (of varying sample sizes), different preprocessing pipelines and software versions, and different statistical frameworks (Castro et al., 2016; Di Biase et al., 2019; Dwyer et al., 2018; Geisler et al., 2015; Madre et al., 2019; Sui et al., 2015; van Haren et al., 2011). When reviewing the literature of studies on SZ vs. HC group difference testing, we did not find any study that performed univariate t-testing and multiple comparison correction at the ROI-level or network-level, rather most work used statistical tests and multiple comparison correction at the voxel-level or edge-level. Combined with the known patterns of heterogeneity present in schizophrenia disorder (Lv et al., 2020; Wolfers et al., 2018), it is unsurprising that our results differ from past studies.”

The choice of a linear kernel for the SVM classification task is not clear to me. Since the input vectors are low dimensional, would a non-linear (e.g. rbf) kernel not perform better?

The choice of a linear kernel was rather arbitrary, as we were following along with an example provided on the scikit-learn website of running SVC and evaluating/plotting the ROC within a cross-validation framework, as shown here: https://scikitlearn.org/stable/auto_examples/model_selection/plot_roc_crossval.html#sphx-glr-autoexamples-model-selection-plot-roc-crossval-py. We chose to use the same hyperparameter settings that were used in this example. We have re-run the classification task using the rbf kernel instead of the linear kernel in the SVC. The results were approximately similar for the deviation score and raw models (both imaging modalities). Therefore, we chose to leave the results as is within the manuscript and added to the manuscript an explanation of the hyperparameter choices as following the example provided by scikit-learn. We have also added to the discussion paragraph on future ideas, that hyperparameter tuning and implementing other algorithms may improve the performance.

“These hyperparameters were chosen based on following an example of SVC provided by scikit-learn.”

“Future research directions are likely to include: … (3) replication and refinement of the proposed benchmarking tasks in other datasets including hyperparameter tuning and different algorithm implementations”

Is there any expected reason why classification in the case of the raw data models would work on the functional but not on the structural measures? (Generally, I would expect a higher noise level in the functional data and thus a more challenging classification scenario.)

Our speculation about the cause of this result is that there may be an effect of confounding variable(s) that the normative models help “clean up” and that the structural data is more effected by this than the functional data. This would also explain why there is such an improvement in the classification when using the structural deviation scores compared to the raw structural features. In the functional models, there is not much of a difference between the deviation score and raw models. While we residualized the raw data of age, sex, and motion (for the functional data) before inputting them into the benchmarking tasks, there could still be some confounding effects present in the raw data. Beyond this, we are unsure and cannot say much regarding the causal nature of this result. We leave this up to future work, and mention this in the Discussion section.

In line 415ff, you mention that normative models allow for the interrogation of patterns in classified vs misclassified samples. Have you seen any such patterns in the schizophrenia vs control benchmark test? (If so, I think it would be instructive for readers to include a brief discussion of observed patterns for the schizophrenia vs controls application.)

This idea was mentioned in the Discussion section in the paragraph on future work. We agree with the reviewer that this is an important and exciting next step to pursue but doing so properly requires significant additional analyses. We believe including these additional analyses in the methods, results, and discussion of the current work would make the paper too crowded.

Reviewer #3 (Recommendations for the authors):I believe this manuscript is in an advanced maturity status; therefore, my recommendations are not particularly major. In no particular order:– Colab examples and tutorials: for some reason, the pnctoolkit package cannot be installed due to dependencies issues. Unfortunately, that dissuaded me from executing the examples (which I was really looking forward to doing and, I believe, is a significant aspect of the overall work).

We have fixed this issue and the Colab notebooks now run as intended. We have also released a version of our code and data (and all required python packages) in a Docker container. However, we note that only the HCP and COBRE datasets have open sharing allowed (not the UMich data). Therefore, the code will not exactly recreate the task 1 and 2 results because we can only partially share the data needed to re-run the analysis. The notebooks that are hosted on GitHub have been shared with the cell outputs saved so that the results and figures can be seen from using all of the data (both COBRE and UMich samples). We have also built a website (no code required) for transferring the pre-trained normative models to new data that is available here (https://pcnportal.dccn.nl/).

– Abstract. The sentence "Across all benchmarks, we confirm the advantage (i.e., stronger effect sizes, more accurate classification, and prediction) of using normative modeling features" is bordering the overstatement. I would suggest bringing up more specific aspects of the results (more quantitative statements, if you will).

We have toned down the message in the abstract and clarified the main results.

“Across all benchmarks, we show the advantage of using normative modeling features, with the strongest statistically significant results demonstrated in the group difference testing and classification tasks.”

– Introduction (ll. 63--72): this paragraph feels a bit out of place. I think the intent is to connect the last sentence (69--72), which I agree is relevant. However, I am not convinced by the flow of the paragraph (e.g., not sure the references to Gau et al. (2021) and Levitis et al. (2021) are a good fit here, as opposed to a discussion). I would invite the authors to rethink how this paragraph is carried out.

We considered moving this paragraph to the discussion but have decided that the current placement is meant to show the audience that beyond the applications of normative modeling (clinical and psychology as mentioned in the paragraph immediately before), there is also active technical development of the normative modeling framework and there are many available open science resources to help users get started with using normative modeling. We feel that this message is best conveyed in the introduction.

– Table 1. HCP's sex ratios. I believe it should say 46.6% of males.

We have corrected this mistake.

– Figure 1. I suggest annotating the different tasks within (C) and (D), and updating the caption accordingly.

We have updated Figure 1 with these suggestions.

– Methods (l. 203). For ignorant readers like this reviewer, a reduced mention of why this particular test (Mann-Whitney U-test) was chosen would be appreciated. Perhaps some further details could also be given in the appendix.

We have added the following to the manuscript to explain why the Utest was used:

“The U-test was used to test for group differences in the count data (of extreme deviations) because the distribution of count data is skewed (non-Gaussian) which the Utest is designed to account for.” https://www.statstest.com/mann-whitney-u-test/

– Methods (l. 212). Some elaboration on how the hyperparameters were selected (i.e., kernel and C constant). I have looked into the corresponding jupyter notebook (which was of course, available, thanks!), and it seems the defaults were used. Could this have an impact on the advantages of normative modeling?

Please see our response to reviewer 2 comment 7 above for an explanation of the hyperparameter choice.

– Methods (l. 232) Evaluation. I would suggest (i) moving the metrics within the body of each task previously (i.e.,) the definition of $\theta$ in equations. (1, 2, 3); and (ii) keeping a subsection with ll. 244--254 with perhaps a more detailed section title (e.g., statistical inference --or evaluation-- of model comparisons).

We have re-formatted the methods to match this suggestion.

– Results section: I would suggest the authors provide subsection titles that advance the contents of the subsection. E.g., in line 271, something along the lines of "Normative modeling showcased larger effect sizes in mass univariate group differences."

We have updated the result subsections as follows:

Normative Modeling Shows Larger Effect Sizes in Mass Univariate Group Differences

Normative Modeling Shows Highest Classification Performance Using Cortical Thickness Normative Modeling Shows Modest Performance Improvement in Predicting Cognition

– ll. 290--292. I suggest reflecting this in Figure 4A, showcasing the corresponding results without multiple comparisons correction (which should be clearly indicated) to give further support to the idea that they were along the same lines.

We considered including the null results on the raw data into Figure 4, however, because we are displaying the t-statistic we ultimately decided it would be misleading to show uncorrected t-statistic maps. The images shown in Figure 4A are showing only the models that survived multiple comparison correction. We have made this more apparent in the cortical thickness map by making the regions that did not survive correction be colorless. We also updated the within 4A figure caption to highlight that there are no significant group differences in the raw maps. We have added to the supplementary files (3 and 4) for all models (deviation and raw, structural, and functional) that contain the group difference test statistics including the t-stat, p-value, and FDR corrected p-value for each ROI.

“For full statistics including the corrected and uncorrected p-values and test-statistic of every ROI, see supplementary files 3 and 4.”

– ll. 285--286. "The lack of any group differences in the raw data was initially a puzzling finding…" Can this be interpreted as normative models being more powered in the discussion? This is sideways mentioned when indicating that normative models have a harmonization effect on the individual metrics, more effective than regressing out obvious confounders. A more frontal/intentional incursion into this discussion would be appreciated.

Please see our detailed response to Reviewer 2 Comment 3. We have added this paragraph to the group difference Results section.

“While there have been reported group differences between controls and schizophrenia in cortical thickness and resting state brain networks in the literature, these studies have used different data sets (of varying sample sizes), different preprocessing pipelines and software versions, and different statistical frameworks (Castro et al., 2016; Di Biase et al., 2019; Dwyer et al., 2018; Geisler et al., 2015; Madre et al., 2019; Sui et al., 2015; van Haren et al., 2011). When reviewing the literature of studies on SZ vs. HC group difference testing, we did not find any study that performed univariate t-testing and multiple comparison correction at the ROI-level or network-level, rather most work used statistical tests and multiple comparison correction at the voxel-level or edge-level. Combined with the known patterns of heterogeneity present in schizophrenia disorder (Lv et al., 2020; Wolfers et al., 2018), it is unsurprising that our results differ from past studies.”

– l. 301. The subsection hierarchy seems broken. It is confusing that this particular result is not at the same level as the other (e.g., l. 311).

We have fixed this formatting.

– l. 365. That "from this work" seemed out of place.

We have reordered the phrasing to say:

“Rather, our results from this work (especially in the comparisons of group difference maps to individual difference maps (Figure 4)) show that the outputs of normative modeling can be used to validate, refine, and further understand some of the inconsistencies in previous findings from case-control literature.”